# Cellular Factors Involved in the Hepatitis D Virus Life Cycle

**DOI:** 10.3390/v15081687

**Published:** 2023-08-03

**Authors:** Keerthihan Thiyagarajah, Michael Basic, Eberhard Hildt

**Affiliations:** Paul-Ehrlich-Institute, Department of Virology, D-63225 Langen, Germany; keerthihan.thiyagarajah@pei.de (K.T.); michael.basic@pei.de (M.B.)

**Keywords:** HDV, HBV, replication, morphogenesis, RNA genome, surface protein, MVB, NTCP

## Abstract

Hepatitis D virus (HDV) is a defective RNA virus with a negative-strand RNA genome encompassing less than 1700 nucleotides. The HDV genome encodes only for one protein, the hepatitis delta antigen (HDAg), which exists in two forms acting as nucleoproteins. HDV depends on the envelope proteins of the hepatitis B virus as a helper virus for packaging its ribonucleoprotein complex (RNP). HDV is considered the causative agent for the most severe form of viral hepatitis leading to liver fibrosis/cirrhosis and hepatocellular carcinoma. Many steps of the life cycle of HDV are still enigmatic. This review gives an overview of the complete life cycle of HDV and identifies gaps in knowledge. The focus is on the description of cellular factors being involved in the life cycle of HDV and the deregulation of cellular pathways by HDV with respect to their relevance for viral replication, morphogenesis and HDV-associated pathogenesis. Moreover, recent progress in antiviral strategies targeting cellular structures is summarized in this article.

## 1. Introduction

Hepatitis D virus (HDV) is a defective RNA virus, inducing the most severe form of viral hepatitis leading to the rapid progression of liver cirrhosis and hepatocellular carcinoma (HCC) [1,2]. It consists of a negative-strand RNA genome composed of 1672–1697 nucleotides depending on the genotype. Thus, HDV is considered the smallest human pathogen [3,4,5,6]. The HDV genome encodes only for one protein, the hepatitis delta antigen (HDAg), which exists in two isoforms termed small (S)- and large (L)-HDAg. These HDAg isoforms mediate genome replication and virion assembly by acting as nucleoproteins [7,8,9]. Hence, HDV is dependent on the envelope proteins of a helper virus for packaging its ribonucleoprotein complex (RNP). Thus, during the natural course of infection, HDV only occurs in coinfection with the hepatitis B virus (HBV), which acts as the helper virus, providing its envelope proteins [10]. Since the envelope proteins are relevant for host tropism, it is assumed that HDV binding and entry to hepatocytes closely resemble HBV. Accordingly, entry inhibitors of HBV also inhibit HDV entry into hepatocytes. In line with that, the most advanced drug against HDV infection is bulevirtide, a synthetic peptide that competitively blocks the hepatic viral receptor of HBV and HDV and thus consequently inhibits virus entry [11]. Moreover, vaccines eliciting an immune response against HBV envelope proteins also provide immunity against HDV. This might explain the decreased HDV prevalence since the beginning of the HBV vaccination campaign. Hence, HDV is mostly prevalent in regions with restricted access to HBV vaccines [12,13]. According to a recent meta-analysis, an estimated 12 million people worldwide have experienced HDV infection, indicating a global prevalence of 0.16% [14]. Another recent meta-analysis estimated an even higher prevalence of 0.8%, approximately 60 million HDV-infected individuals [15]. Nevertheless, HDV prevalence varies significantly within geographical regions. In Europe, the HDV prevalence is estimated to be 0.05%, whereas in African regions, the prevalence is over 0.3%. Mongolia has the highest national HDV prevalence with 4% in the general population and 36.9 % in the HBsAg-positive population. In comparison, the global HDV prevalence amongst HBsAg-positive individuals was estimated to be 4.5% [14].

## 2. HDV—The Outsider

HDV is a unique human pathogen. Its structure and mechanism of replication are similar to those of viroids, infectious subviral plant pathogens. However, unlike viroids, which contain a very small non-encoding RNA genome of only a few hundred nucleotides (nt), HDV harbors a 1672–1697 nt RNA genome encoding for two functional protein isoforms. Moreover, unlike viroids, it is dependent on a helper virus for envelopment. Due to these very unique traits of HDV, defining it either as a virus or subviral agent is ambiguous [3,4,5,6,16]. Because of sequence variations of up to 20%, HDV is divided into eight distinct genotypes and several subgenotypes. Amongst them, genotype 1 is the only globally scattered HDV variant, whereas all other genotypes are prevalent in specific regions. For instance, genotype 2 is prevalent in Southeast Asia, whereas genotypes 5 to 8 are prevalent in African countries [17]. Interestingly, the genotype seems to influence the clinical outcome. A retrospective study by Spaan et al., for instance, indicated better prognosis and fewer episodes of hepatic decompensation in patients infected with genotype 5 compared to patients infected with genotype 1 [18]. Differences in replication and infection efficacies between genotypes as observed in in vitro studies of Wang et al. might explain these discrepancies in pathogenicity [19].

Recently, delta-like agents infecting numerous non-human hosts such as birds, fish, rats and woodchucks, among others, have been discovered [20,21,22]. Interestingly, these delta-like agents are not necessarily associated with hepadnavirus coinfections and mostly share minimal homology to HDV [21,23].

## 3. HDV Structure

The HDV genome consists of a covalently closed, single-stranded RNA with negative polarity. Interestingly, the genome and antigenome both have a high rate of self-complementarity (>74%), which results in the folding of the RNA molecules into an unbranched, rod-like structure resembling double-stranded RNA [3,4]. The RNA genome and antigenome encode ribozymes, intrinsic self-cleaving RNA motifs [24]. Interestingly, these HDV ribozymes have rapid self-cleaving rates without the need for protein factors or upstream sequence requirements and therefore are widely used as biological tools [25,26,27]. Moreover, the HDV genome also encodes two protein isoforms, termed small (S)- and large (L)- Delta antigen (HDAg). L-HDAg results from RNA-editing-mediated read-through of the S-HDAg sequence and thus shares the same N-terminus as S-HDAg with an extended C-terminal region. Both isoforms have distinct and partially opposing functions in the viral life cycle [7,8,9], which will be discussed in detail later on (Section 6, Section 7 and Section 8). Furthermore, both HDAgs have distinct functional domains, including an arginine-rich RNA-binding domain [28,29], a coiled-coil sequence mediating oligomerization [30] and a nuclear localization signal for nuclear import [31]. L-HDAg has an additional nuclear export signal as well as a farnesylation site in its C-terminal extension [32,33,34,35]. The RNP complex of HDV is composed of large S- and L-HDAg multimers, which are bound to the RNA genome, possibly by assembling into nucleosome-like structures [36,37]. The approximately 18 nm sized RNP complexes are enveloped by HBV HBsAg containing host-derived lipid bilayers. The resulting enveloped virus has a diameter of 35 nm–37 nm [38,39] (Figure 1). 

## 4. HBV Structure and Replication Cycle

HBV encodes three envelope proteins, namely small (S)-, middle (M)- and large (L)-HBsAg, which are encoded by a single open reading frame (ORF) and are regulated by two upstream promoters, pre-S1 and S. Thus, all HBsAg share the same C-terminus with extended N-termini. The pre-S1 promoter regulates the transcription of a 2.4 kb mRNA transcript, which is translated into LHBs, whereas the S promoter regulates the transcription of a 2.1 kb mRNA, which in turn is translated to M- and SHBs. Moreover, the HBV genome consists of a core promoter, which regulates the expression of the HBV core protein, HBcAg. HBcAg encapsidates the HBV genome. The core promoter also regulates the expression of a precursor protein, termed precore protein p25, which has a 29 amino acid N-terminal extension relative to HBcAg. This precursor is post-translationally modified by cleavage at the N- and C-terminus giving rise to a non-particulate protein termed HBeAg. Due to its distinct N- and C-terminus, HBeAg is secreted as a monomer and carries out regulatory and immune-modulatory functions [40,41,42,43]. Interestingly, partial cleavage of p25 gives rise to another precore protein termed p22cr. This protein lacks the arginine-rich domain, which mediates HBV capsid and genome interaction and therefore is found in Dane-like particles lacking viral DNA [40,44]. Since HBcAg, HBeAg and p22cr share the same constitutive 149 amino acid sequence, they can bind the same monoclonal antibodies and are thus referred to as hepatitis B core-related antigens (HBcrAg). Serum HBcArg is used as a surrogate marker for intrahepatic HBV DNA in infected patients with undetectable HBV DNA levels [45]. Furthermore, the HBV genome also encodes the HBx protein, which is a viral regulatory protein with pleiotropic functions [46]. Additionally, the HBV genome harbors two enhancer elements upstream of the core and X promoter, respectively. These enhancer elements further modulate HBV replication [47,48,49]. During the replication of HBV, the viral genome is converted to a covalently closed circular DNA (cccDNA), which in turn acts as the template for the transcription of the 3.5 kb viral precursor RNA, termed pregenomic RNA (pgRNA), as well as for the 2.4, 2.1 and 0.7 kb subgenomic viral RNAs encoding for the HBsAg variants and the HBx protein. Interestingly, different types of particles egress during active HBV replication, including, amongst others, infectious HBV virions containing encapsidated viral DNA, as well as non-infectious subviral particles (SVPs) lacking a genome and capsid. Furthermore, non-enveloped nucleocapsids (naked capsids), the abovementioned Dane-like particles containing capsids but lacking a viral genome and RNA containing virions were also described in vivo. However, these types of particles are only hardly detected in the bloodstream of infected individuals, unlike SVPs, which are the most abundant particles in infected patient sera. SVPs are either spherical or filamentous. Spherical SVPs are released by the secretory pathway, whereas filamentous SVPs utilize the ESCRT/MVB pathway like infectious HBV virions. The different types of particles are characterized by different ratios of L-, S- and MHBs. Spherical SVPs, which are the most abundant type in HBV-infected patient sera, are composed mostly of SHBs and almost no LHBs [50,51,52]. Interestingly, the HDV envelope resembles spherical SVPs. It consists mostly of SHBs and little M- and LHBs [10].

## 5. Simultaneous Coinfection vs. Superinfection of HBV and HDV

Studies by Perez-Vargas et al. in 2019 described that HBV-unrelated viruses such as flavivirus and vesiculovirus can also act as helper viruses by providing their envelope proteins for HDV RNP packaging. According to their study, hepatitis C virus (HCV) can propagate HDV infection in humanized mice for several months [53]. However, a Germany-based study of over 300 HCV-infected patients found no solely HCV and HDV-positive patients [54]. Another clinical study based on 160 HCV-positive patients originating from a population with high HDV prevalence found two HDV, HCV-positive and allegedly HBV-negative patients. However, an occult HBV infection in these patients could not be ruled out, thus emphasizing a low probability of HDV coinfection with other HBV-unrelated viruses in vivo [55].

There are two different modes of HDV/HBV infection, termed co- and superinfection. Roughly speaking, coinfection is defined as the simultaneous acute infection of HDV and HBV, whereas superinfection is defined as an HDV infection of an individual with a pre-existing chronic HBV infection. Interestingly, both modes differ in terms of severity and clinical outcome. For instance, an HDV/HBV coinfection in healthy adults is mostly transient, since it is eliminated by the host immune system. Only in 5% of the coinfected cases does HDV infection become chronic. In contrast to that, superinfection leads to chronicity in 80% of the cases. In general, chronic hepatitis D is characterized by a rapid progression of liver cirrhosis and HCC. The risk of developing liver cirrhosis is two times higher in chronic HDV-infected compared to chronic HBV monoinfected individuals [56,57,58]. Studies by Rizetto et al. reported that 70% of chronic hepatitis D patients developed cirrhosis within 5 years [59]. Moreover, Rosina et al. estimated a probability of survival for patients with HDV-induced cirrhosis of only 49% within 5 years. Therefore, chronic HDV infection is considered the most severe form of all chronic viral hepatitis [56,60,61,62]. Although liver damage induced by chronic HDV infection is assumed to be mostly immune-mediated [56], the capability of HDV to regulate and interact with a vast amount of hepatocellular host factors in favor of its own replication emphasizes the cytotoxicity and oncogenicity of HDV. Therefore, in this review, we describe cellular host factors that are involved in the HDV replication cycle (Section 6, Section 7 and Section 8) and discuss the contribution of the involved host cell factors in HDV-induced pathogenesis (Section 12). From now on, both HBV/HDV infection modes will be addressed as coinfection for the sake of convenience. 

## 6. Host Factors Involved in HDV Entry

HDV shares surface proteins with HBV. Therefore, HDV is frequently considered a surrogate model to study the HBV entry process. However, it should be considered that the interaction with the HBV capsid on the one hand and with the HDAg on the other hand impacts the conformation of HBsAg and thereby could affect the interaction with entry factors and the entry process (Figure 2). There are two crucial parts mediating HDV/HBV entry, the N-terminal 75aa of the PreS1 domain in its myristoylated form and the antigenic loop formed in the SHBs. There is a low-affinity interaction of the antigenic loop in the S domain with heparin sulfate/glypican. Electrostatic interactions between negatively charged heparin sulfate and R122 and K141 in the antigenic loop of SHBs mediate a low-affinity enrichment of the viral particle on the cell surface, leading to local enrichment [63,64]. This is followed by the binding step to the Na^+^-taurocholate cotransporting polypeptide (NTCP) encoded by the SLC10A1 gene, which acts as a bona fide receptor [65].

Although the X-ray-based structure of human NTCP is not solved, there are well-established concepts about the structure of NTCP based on mutational analyses of NTCP and their impact on NTCP functionality. Moreover, there are bacterial homologs of NTCP (ASBTNM and ASBTYf) whose X-ray structure was solved and robust structural prediction data are available [66]. These structural data are corroborated by recent cryo-eletronmicroscopy data of the human NTCP [67,68,69]. NTCP encompasses nine transmembrane domains (TMDs I–IX). The TMDs I, IV and V form a panel domain. The core domain is formed by the two inverted triple-helix repeats encompassing TMDs II–IV and VII to IX. A large amphipathic cavity is formed by the TMDs VI and IX towards the cytoplasm and laterally to the hydrophobic core of the cell membrane [69]. 

A central function of NTCP is the transport of bile acids across the sinusoidal membrane of the hepatocytes. This is relevant for the regulation of bile salt concentration in the blood. Almost all physiological bile acids can be transported by NTCP. In addition, sulfated steroid hormones and sulfated jodothyronines are NTCP substrates. For a recent overview, see [70].

With respect to the life cycle of HBV and HDV, the identification of NTCP as an entry receptor represented a milestone [65,71,72]. It was found that aas 2–48 of the myristoylated preS1-domain LHBs (myr-preS1 2–48) are absolutely essential for binding of HBV/HDV to NTCP [73,74]. Although a detailed analysis of the LHBs-NTCP interaction is not available so far, it is well established that aas 77–84 in the loop between TMD II and III and aas 157–165 as part of TMD V are essential for the interaction with the receptor-binding domain in LHBs. Moreover, Y146, which is part of the loop connecting TMD IV and V, interacts with the receptor-binding domain [69]. These aa residues that were identified to mediate the binding of NTCP to the receptor-binding domain are not part of the pore in which bile acids bind to NTCP [70,75]. In accordance with this, point mutations affecting the binding to the LHBs-receptor-binding domain do not interfere with the binding/transport of bile acids [68,76]. Based on this, it can be concluded that in principle, an interference with LHBs-receptor-binding domain binding to NTCP is possible without affecting the bile acid transport function of NTCP.

In differentiated hepatocytes, NTCP is present on the basolateral membrane. NTCP homodimerizes. In the homodimer, each subunit is functionally active to serve as a bile acid transporter [71,77]. The residues mediating dimerization so far are not characterized. However, based on deletion analyses, it can be concluded that the C-terminus of NTCP is dispensable for dimerization, while aas 221–240 (harboring a GXXXG dimerization motive) and 271 to 290 are crucial for dimerization [78]. The relevance of NTCP dimerization for the HBV/HDV receptor function is still unclear, but there is evidence that the binding of a receptor-binding domain peptide is reduced by interfering with the NTCP dimerization. 

Troglitazone, an antidiabetic and anti-inflammatory drug, is known to interfere with NTCP oligomerization and exerts an inhibitory effect on HBV entry, suggesting the relevance of NTCP oligomerization for HBV entry [79,80]. However, there are additional targets such as NF-κB signaling, which are affected by Troglitazone. The pronounced hepatotoxicity of this compound argues against further characterization as HBV entry inhibitors, but similar molecules are currently tested with respect to their impact on HBV/HDV entry [81].

Bulevirtide, previously known as Myrcludex B, is at present the only approved drug for HDV treatment. Conditional market authorization by the European Medicines Agency (EMA) was guaranteed in 2020. Bulevirtide is an N-terminally myristoylated peptide encompassing aas 2–48 of the PreS1 domain. It is assumed to compete with the HDV binding to NTCP and thereby prevent the spread of the infection. The impact of Bulevirtide/Myrcludex B on the HDV life cycle has been studied in various studies. There was a good safety profile and a significant impact on virus replication. It is assumed that the regression of HDV RNA in treated patients contributes to the improved clinical outcome of the patient. The combination of Bulevirtide/Myrcludex B with PegIFNa has an additional and synergistic antiviral effect on HDV replication [11,82,83].

Although there are strong preclinical and clinical data describing the mode of action and the clinical impact of Bulevirtide on the HDV life cycle, there are still open questions, i.e., concerning the duration of the treatment’s potential long-term effects due to minor interference with bile acid metabolism and viral relapse after discontinuation of the treatment. Moreover, it would be very helpful to establish predictors for response and define criteria for discontinuation of the therapy [84,85]. New results from a phase III clinical study found that HDV RNA and ALT levels were significantly reduced in chronic HDV patients after 48 weeks of Bulevirtide treatment [85]. Furthermore, the study is designed to assess long-term efficacy as well as sustained virologic response after 144 weeks and 168 weeks, respectively. An additional phase II trial is currently ongoing, investigating a combination therapy of Bulevirtide with PEG-interferon α [86]. 

The epidermal growth factor receptor (EGFR) has been described as a relevant cofactor for the entry of HBV and HDV. NTCP was found to interact with EGFR. For HBV, it was found that inhibition of the NTCP/EGFR interaction abolishes HBV entry, while EGF stimulation favors HBV entry [79,87]. There is evidence that the interaction with EGFR is mediated by the residues aa131–aa150 of NTCP. However, the relevance of EGFR/NTCP interaction for HDV entry so far is not fully clear. It is assumed that the interaction with EGFR or with E-Cadherin is required for the endocytic uptake via clathrin. As this process requires activation/phosphorylation of EGFR, the promoting effect of EGF stimulation on HBV entry could be explained [88]. Moreover, the knockdown of proteins being involved in the EGFR internalization such as adaptor-related protein complex 2 subunit α 1 (AP2A1) and receptor pathway substrate 15 (EPS15) as well as the NTCP cofactor interferon-induced transmembrane protein 3 (IFITM3, which is involved in endosome formation and endosome sorting) indicate the relevance of EGFR-mediated endocytosis for HBV/HDV entry [89,90,91].

A central question for the life cycle of HBV and HDV is the escape from the endosomal compartment. For HBV, it is assumed that the internalization occurs by clathrin-mediated endocytosis. The general mechanism is that fusion of the viral membrane with the endosomal membrane enables escape from the endosomal compartment [87,92,93]. Indeed, there are reports about the identification of fusogenic sequences within the HBsAg that could mediate release from the endosomal compartment [94,95,96,97]. An alternative concept is based on the identification of a cell-permeable peptide, the TLM peptide (translocation motive) within aas 41–52 of the PreS2 domain [98]. The TLM has the capacity to mediate the transfer of peptide proteins or even complete capsids across cellular membranes if it is fused to the cargo protein [99,100]. This makes the TLM an interesting tool enabling protein transfer across cellular membranes. The presence of a TLM peptide is conserved among all hepadnaviruses, suggesting relevance for the HBV life cycle. However, there are conflicting results based on studies with HDV and HBV providing evidence that the TLM peptide is dispensable for the entry process of HBV/HDV [101]. On the other hand, the deletion of the TLM peptide abolishes HBV infectivity and prevents the release of the nucleocapsid from the endosomal compartment, arguing for a central role of the TLM-peptide in the viral life cycle [102].

## 7. HDV Interaction with the Host during Replication

As HDV relies on the host cell for replication and on the envelope proteins of HBV for release, the interaction between HDV and cellular host factors is plentiful. With the limited coding capacity of the HDV genome, interactions are limited to either one of the hepatitis D antigens (S- or L-HDAg) or the RNA genome (gRNA) itself and its respective antigenomic RNA (agRNA), which serves as replication intermediate. Regulation and control of HDV replication requires a delicate interplay of subcellular localization, genome amplification and mRNA synthesis, multiple of which are performed by S-HDAg, which is essential for the replication cycle (Figure 3). 

### 7.1. Interaction of HDAg with Host Factors Relevant for Replication

The HDAg contains multiple distinct regions that regulate its function in the different stages of the HDV life cycle. The N-terminal region harbors an RNA-binding domain (RBD, amino acids (aas) 2–27), followed by the coiled-coil sequence for oligomerization (aas 31–52). The nuclear localization sequence (NLS, aas 68–88) is localized just before the main RBD (aas 97–146) that is flanked by two arginine-rich motifs (ARM, aas 97–107 and aas 136–46). The main interaction of HDAg with RNA occurs via these ARMs, however, the linker region between both ARMs was shown to contribute to the interaction [29]. L-HDAg, which is elongated by about 19 amino acids, holds an additional nuclear export signal (NES) followed by the isoprenylation site at the C-terminal end [33,35,103] (Figure 1). 

Upon transport of gRNA associated with ribonucleoprotein (RNP) consisting of S- and L-HDAg into the nucleus, the RNP is assumed to recruit DNA-dependent RNA polymerase II (RNAP-II) and initiate transcription of S-HDAg mRNA [8]. This initial import into the nucleus is carried out by a part of the nuclear pore complex (karyopherin 2a) binding to the NLS motif of HDAg [104,105,106,107]. Interestingly, it was observed that one ARM is sufficient for HDV RNA import in vivo, introducing a redundancy mechanism to ensure transport into the nucleus in case of steric hindrance or masking of HDAg domains [28,29]. Shuttling of HDAg, gRNA, agRNA and, at later stages, HDV RNPs is essential for the replication and spread of HDV and needs to be ensured for the control of the HDV life cycle. Subsequent association with RNAP-II through binding of S-HDAg and the start of transcription of HDAg mRNA has been suggested. The finding that HDAg mRNA is post-transcriptionally modified with a 5’-7 methylguanosine cap and poly-A tail is consistent with transcripts created by RNAP-II [108]. In vitro transcription assays and the sensitivity of gRNA synthesis to α-amanitin further support the role of RNAP-II in HDV transcription [109,110,111,112]. The heterogenous nuclear ribonucleoprotein C (hnRNPC) is described to form nuclear complexes with primary transcripts from RNAP-II and assist mRNA processing, nuclear export and translation [113,114]. S-HDAg interaction with hnRNPC was observed to occur through their RBDs, with a knockdown leading to a decrease in HDAg expression [115]. This decrease in HDAg expression suggests HDV mRNA as the primary interaction partner since hnRNPC is responsible for the allocation of transcripts to the correct export pathway. Generated HDAg mRNA is exported from the nucleus and translated in the cytoplasm with newly synthesized S-HDAg being relocated into the nucleus. As HDAg mRNA possesses a poly-A tail and L-HDAg has been observed to interact with the nuclear RNA export factor 1 (NXF1) and the adapter protein Aly, one could speculate that both NXF1 and Aly are responsible for the export of HDV mRNA from the nucleus. The recruitment of both NXF1 and Aly as well as other RNA-binding proteins belonging to the transcription export (TREX) complex that could associate either with HDAg or HDV mRNA remains to be identified. The proline-rich NES of L-HDAg interacts with the RNA helicase UAP56, responsible for the recruitment of Aly, which in turn recruits NXF1 as the export receptor, facilitating the export of RNPs [33,116,117]. For a detailed review of mRNA export, see [118,119].

After sufficient synthesis of S-HDAg, gRNA is transported into the nucleolus, where production of the replication intermediate antigenomic RNA takes place [120]. Nucleolin (NCL) and nucleophosmin (B23) are phosphoproteins located inside the nucleolus/nucleus and can interact with HDAg [121]. Both possess multiple functions, one of which is nuclear shuttling and ribosome biogenesis. Upregulation of the expression of nucleophosmin as well as accumulation of nucleolin together with agRNA was observed for HDV [122,123]. The binding site of nucleolin resides at the conserved arginine-rich domain, while nucleophosmin interacts with the NLS of HDAg. Interaction with B23 was observed to occur preferentially with S-HDAg in vivo, while both isoforms could interact with B23 in vitro. Two highly acidic regions of B23 are required for the binding to HDAg. However, unlike other viral proteins binding to B23 via their ARM, B23 did not interact with the ARM of HDAg and could require further conformational assistance for its binding ability [122]. Furthermore, both proteins form complexes together with HDAg localized inside nucleoli, but the exact function of these proteins in HDV replication has not yet been determined. 

The production of the replication intermediate inside the nucleolus is attributed largely to RNAP-I, as agRNA synthesis is resistant towards α-amanitin, an RNAP-II inhibitor, treatment [112,124]. However, a direct interaction between S-HDAg and RNAP-I could not be observed apart from an association with transcription factor SL1, which belongs to the initiation complex responsible for the recruitment of RNAP-I to the promotor of ribosomal RNA genes [125]. Additionally, the binding of HDV RNA to RNAP-III was observed, but the exact function of this interaction remains unclear [109,124,126]. Nonetheless, replication from gRNA to agRNA is proposed to follow a rolling circle mechanism similar to the replication of plant viroids [127,128]. As such, concatemeric RNA molecules containing multiple full-length agRNAs or gRNAs are created [129]. Subsequent conversion of these multimeric molecules to monomeric gRNA or agRNA is carried out by the viral ribozymes localized on both gRNA and agRNA [24]. Self-cleavage by the cis-ribozyme is followed by host-aided ligation to generate covalently closed circular RNA molecules [129,130]. Afterwards, monomers of agRNA are shuttled back into the nucleoplasm from nucleoli where synthesis of the respective gRNA can take place in a similar manner through a rolling circle mechanism including self-cleavage and ligation. The switch between mRNA and gRNA synthesis most likely occurs via the phosphorylation status of S-HDAg. Serine at position 177 (S177) was reported to dictate the interaction with hyper- and hypophosphorylated RNAP-II in dependence on its phosphorylation status [131,132,133]. In that regard, phosphorylation of S177 favors the interaction with hyperphosphorylated RNAP-II, while unphosphorylated S177 primarily interacts with hypophosphorylated RNAP-II. As the phosphorylation status of RNAP-II determines the function such as early binding to promotor regions, elongation and termination of transcription, specific recruitment of the hyperphosphorylated RNAP-II might thus initiate replication via the agRNA intermediates [131,134,135,136]. The export of newly synthesized gRNA is facilitated by the large HDAg, which appears only in the later stages of infection with remnants being detectable from the initial RNP complexes. Expression of L-HDAg is strictly regulated and repressed by S-HDAg via inhibition of required interactions with adenosine deaminase acting on RNA (ADAR1), highlighting the regulatory function of S-HDAg [137,138,139]. ADAR1 edits the termination codon UAG belonging to the S-HDAg mRNA, transforming adenine into inosine and leading to the elongation of the reading frame and ultimately the formation of L-HDAg. The site of ADAR1 editing has been termed an amber/W site, as editing leads from an amber stop codon to a tryptophan (W) codon. Interestingly, the substrate for this editing step is not the mRNA but rather the agRNA. This in turn leads to the formation of gRNA templates carrying the modified reading frame and, over time, leads to the accumulation of edited gRNAs as long as replication occurs. Without any regulation of this editing process, replication would ultimately come to a halt, and edited gRNAs, which are replication-deficient as the de novo synthesis of S-HDAg would not be possible, would be packaged and released instead of replication-competent viroids. Transfection of gRNA carrying the L-HDAg ORF into S-HDAg producing TSδ3 cells did not lead to replication of edited RNA [140]. The rate of agRNA editing strongly depends on the available structure of the RNA as well as the association rate of HDAg, which is capable of blocking the interaction of ADAR1 with the editing site. Editing rates by ADAR1 were observed to be different between genotypes of HDV based on the degree of base pairing of their RNA in the area of the amber/W site [141,142,143]. However, the exact areas of HDV RNA required for interaction with and editing by ADAR1 are yet to be elucidated. Interestingly, the amber/W site is edited with much greater probability when compared to other possible targets in HDV RNA [138]. This might be facilitated by the secondary structure of the HDV RNA such as the frequency of G-A dinucleotides (5’ neighbor nucleotide preference, where G is the least favored and U the most) [144] and the rate of disruption of the secondary RNA structure by bulges and loops [145]. Furthermore, HDAg was observed to suppress editing of the amber/W site, therefore inherently limiting the amount of edited genomes present at one time independent of their current amounts [138,139]. This represses the amount of editing events and limits the available L-HDAg while allowing for further, albeit reduced, replication of unedited genomes. Once a certain equilibrium between L- and S-HDAg has been reached inside the nucleus, L-HDAg begins to drive the egress of gRNA from the nucleus in the form of newly formed genomic ribonucleoprotein (RNP) complexes [33,146]. The added C-terminal domain of L-HDAg contains a farnesylation signal, which is recognized by the host’s farnesyl transferase (FTase) and transfers a covalently bound farnesyl residue to cysteine 211 (C211) [35,147]. Farnesylation of L-HDAg is an essential step for the interaction between the HDV RNP and the HBsAg envelope and subsequent release but additionally was described to lead to repression of HDV transcription and favor viral packaging [34,148]. As such, the inhibition of FTase by Lonafarnib, a farnesyltransferase inhibitor currently in clinical phase III, leads to an accumulation of intracellular HDV RNA and blocks the interaction between L-HDAg and surface antigens of HBV [147,149]. Blocking of farnesylation and therefore assembly causes accumulation of L-HDAg, which in turn represses replication through its presence as well as the release of progeny virus [139,150,151]. Furthermore, a recent study found that Lonafarnib treatment facilitated a shift in the amount of edited/unedited HDV genomes towards the edited version, causing a loss of infectivity [152]. The specific modification of farnesylation further leads to conformational changes in L-HDAg, probably contributing to the difference in functions. Proline at position 205 was found to be indispensable for HDV particle formation and function of NES [33]. Export from the nucleus was observed to occur in a chromosome region maintenance 1 (CRM1)-independent manner [153]. The NES-interacting protein (NESI) was found to be associated with L-HDAg via its C-terminal end and in a complex with lamin A/C to facilitate nuclear export. 

Aside from the already mentioned post-translational modifications (PTM), further control of replication occurs via methylation and acetylation of S-HDAg [154,155,156]. Protein arginine methyltransferase 1 (PRMT1) is capable of methylating arginine at position 13 located inside an RGGR motif in the N-terminal RBD, which was found to be essential for the replication of agRNA to gRNA [154]. A similar effect on gRNA synthesis was observed regarding acetylation, which was found to be relevant for lysine at position 72 [156]. Acetylation is carried out by the histone acetyltransferase (HAT) domain of p300, a cellular acetyltransferase. Both methylation and acetylation cause a significant shift in the subcellular localization of HDAg to favor a cytoplasmic distribution. Additionally, small ubiquitin-related modifier isoform 1 (SUMO1) was found to interact and modify S-HDAg by sumoylation at lysine residues, leading to an increase in gRNA and mRNA synthesis [157]. Whether this process of upregulation is facilitated by a change in localization, protein–protein interaction, protein-RNA interaction or stability of S-HDAg through sumoylation remains to be elucidated. Even though both S- and L-HDAg carry identical PTM sites, phosphorylation of S-HDAg occurs at serine/threonine, while only serine residues are phosphorylated in the case of L-HDAg [158,159]. Phosphorylation of HDAg was observed to be carried out by casein kinase II (CK II, S2 and S123), double-stranded RNA-activated kinase R (PKR, S177, S180 and T182), extracellular signal-related kinases 1 and 2 (ERK1/2, S177) and protein kinase C (PKC, S210) [159,160,161]. Furthermore, the phosphorylation level of L-HDAg was found to be at least sixfold higher than that of the S isoform [162,163]. This observation of hyperphosporylation could largely be attributed to the differing protein conformation of S- and L-HDAg and not necessarily be based on farnesylation [148,163,164]. Aside from the already mentioned S177, S2 and S123 are two putative phosphorylation sites for interaction with CK II. Inhibition of CK II led to a shift in L-HDAg localization into SC35 domains (nuclear speckles) [160,165]. Interestingly, it was observed that the phosphorylation status of L-HDAg does not affect the formation of viral RNPs [166]. A later study proposed a more specific mechanism involving the phosphorylation status of S123. Synthesized L-HDAg would be transported into the nucleus via its NLS, followed by later dephosphorylation of S123 leading to export from the nucleolus into SC35 domains where L-HDAg could again be phosphorylated as well as prenylated, leading to conformational changes exposing the NES for nuclear export and subsequent release [165]. Similarly, CK II has been observed to be relevant for the replication of HBV by affecting the polymerase function. Inhibition of CK II led to the abolishment of HBV replication during the early stages of infection [167]. To make things more complicated, mutation of C211 has been observed to prevent phosphorylation altogether, highlighting the significant role of farnesylation of L-HDAg and its regulation of PTMs [168]. However, Choi et al. observed no change in phosphorylation and attributed this discrepancy to conformational changes due to the additional C-terminal extension rather than farnesylation [163]. Lastly, S- but not L-HDAg was shown to interact with the histone linker H1e by the N-terminal domain and affect replication [169]. The role of H1e in chromatin organization would further support the proposed histone-like function of HDAg [37]. 

The only viral protein of HDV possesses a plethora of functions and is responsible and essential in all steps of the viral life cycle. Both isoforms of HDAg are associated with different functions and determine the stage of the viral life cycle. However, the identification of specific PTM patterns, their specific functions and requirements for the interaction with the relevant host factor or factors as well as the guiding role depending on the stage of the viral life cycle requires further studies.

### 7.2. Interaction between HDV RNA and the Host

The secondary structure of HDV RNA makes it an attractive target for RNA-binding proteins and host factors. As such, glyceraldehyde 3-phosphate dehydrogenase (GAPDH) was reported to enhance ribozyme activity when interacting with agRNA [170]. Observation of the interaction between gRNA and GAPDH could additionally facilitate the shuttling of GAPDH into the nucleus at the early stages of infection where it would later increase the rate of cis-cleavage during the rolling circle replication [170,171,172]. Furthermore, the interaction and recruitment of HDV RNA with different kinases that facilitate phosphorylation could be relevant for the regulation of HDAg functions. Double-stranded RNA-activated protein kinase R (PKR) was observed to interact with all forms of HDV RNA leading to the activation of PKR and therefore could represent a mechanism for recruitment of PKR towards RNPs, facilitating the phosphorylation of HDAg [161,173,174]. HDV RNA has been observed to interact with polypyrimidine tract-binding protein via PTB-associated splicing factor (PSF) and 54 kDa nuclear RNA-binding protein (p54nrb), which have been observed to play a potential role in the directing of RNAP-II to HDV RNA, which would support and supplement replication [175,176]. Especially PSF was found to be capable of interacting with HDV RNA and RNAP-II simultaneously while binding in the proposed promotor region/stem loop. Furthermore, PSF has been described to be involved in splicing, regulation of transcription, polyadenylation and nuclear shuttling and might therefore be involved in multiple steps within the HDV life cycle [177]. Alternative splicing factor/splicing factor 2 (ASF1/SF2) has been described to be involved in early mRNA processing as well as in association with viral replication in the context of other viruses [178]. However, while ASF/SF2 was found to interact with HDV RNA, its binding site was determined to be outside of replication-relevant areas [171,179]. It is assumed that PTMs of ASF/SF2 could be required for an as-of-yet-undiscovered function in HDV biology. Similarly, the eukaryotic translation elongation factor 1 alpha 1 (eEF1A1) was described to be involved in the life cycle of HDV because of its association with HDV RNA. Its usual role in RNA translation by delivering tRNAs to the ribosome might suggest a supporting role by either enhancing protein synthesis or through interaction with RNAP-II, enhancing replication [171]. Another splicing factor that was shown to interact with gRNA is splicing factor 3 B155 (SF3B155). In this context, HDV promoted changes in the splicing pattern of RBM5, a tumor suppressor, leading to a reduction in protein levels and therefore promoting disease progression/tumorigenesis [180]. Additionally, paraspeckle protein 1 (PSP1) was found to be delocalized during RNA replication as well as to interact with HDV RNA [109,110,175,176,181,182,183]. PSF, p54nrb and PSP1 are proteins localized in SC35 domains and are involved in RNA transport and stabilization among others [184]. 

The interaction between nuclear host proteins and HDV RNA is plentiful and might aid in viral propagation or pathogenic effects such as tumorigenesis. Further characterization of HDV-host interactions will allow for a better understanding of the mechanisms causative to disease progression and the development of antiviral strategies. Especially insights into the regulation of HDV RNA replication and its shuttling provide valuable information. 

## 8. Host Factors Involved in HDV Release

While many aspects of the release of HBV are characterized, the release of HDV is much less studied. As HBsAg as surface proteins are common to both HBV and HDV, it is assumed that the release of both viruses shares many aspects (Figure 2). HDV replication and release can persist in the absence of HBV replication. The production of the HBV surface proteins is sufficient [185,186]. However, there are little experimental data characterizing in detail the release pathways of HDV, and it remains unclear if HDV is released via the route used by spheres or via the pathway used by filaments and the infectious HBV particles [19,187,188]. In addition to infectious HBV particles, naked capsids and subviral particles are released from HBV-expressing cells [51,189]. While infectious viral particles are characterized by the presence of the genome, subviral particles in the form of spheres and filaments are characterized by a lack of core protein and viral genomes. Selective overproduction of SHBs allows the formation and secretion of spheres characterized by a diameter of 22 nm and a highly ordered octahedral structure. In contrast to this, patient-derived spheres are less regularly organized [190,191,192].

The spheres found in patient-derived sera are almost exclusively formed by SHBs and contain variable but small amounts of MHBs and LHBs. The assembly of spheres starts in the endoplasmic reticulum (ER) with the formation of SHBs dimers. The dimers are transported to a pre-Golgi compartment to complete the assembly process. In the ERGIC, a variety of structural changes of the assembled HBsAg dimers occur, transitioning from a more filamentous structure to spheres. After the formation of the spheres, they are transported via the classic constitutive secretory pathway through the Golgi complex. Here, N-glyosylation in the form of high mannose carbohydrates occurs at N-146. The integrity of this glycosylation site is crucial for the release of spheres [51,193].

It has been assumed that subviral particles in the form of spheres and filaments and virions all are released via the constitutive secretory pathway. More recent data revealed that the release of virions and filaments differs from the secretion of spheres [191]. 

For HBV, it was found that the release of the virions depends on the integrity of the ESCRT (endosomal sorting complex required for transport) machinery and occurs via multivesicular bodies (MVBs). MVBs represent a kind of track switch enabling further steps towards lysosomal degradation or fusion with the plasma membrane and subsequent release in the form of extracellular vesicles designated as exosomes. It is assumed that ESCRT-I, -II and -III complexes together with Vps4 ATPase and other associated proteins are involved in the MVB-dependent release of HBV. The interaction between the different components occurs in an ordered sequence. For a variety of viruses, interference with the ESCRT machinery was described. The interaction with the ESCRT machinery depends on structural prerequisites defined as late (assembly) domains [189,194,195,196,197].

Three distinct classes of late assembly domains participating in virus budding are known: YPXL late domains interact with ALIX, an ESCRT-I and -III interaction partner; P(T/S)AP late domains bind to TSG101, a subunit of ESCRT-I; and PPXY late domains bind to NEDD4 family proteins [198,199]. In contrast to other viruses, which are released via MVBs, the surface protein of HBV lacks a classic late domain as described above. However, it was found that the release of HBV depends on α-taxilin, which binds to the N-terminal part of the PreS1 domain. Interestingly, α-taxilin contains a YXXL-motive in the form of a YAEL-sequence resembling the late domain described for the Gag protein of equine infectious animia virus (EIAV) [200]. In HBV-expressing cells, the expression and amount of α-taxilin are strongly enhanced. Silencing of α-taxilin expression inhibits the release of HBV virions, while the release of spheres is not affected. The interaction of LHBs with α-taxilin allows HBV to overcome the lack of a classic late domain within the surface proteins, which is a prerequisite for the interaction with the ESCRT machinery [198,199,201]. The late domain within α-taxilin in the form of the YAEL sequence interacts with the ESCRT-component tsg101. Thus, by the binding of α-taxilin to the N-terminal part of the PreS1 domain of LHBs, the lack of a late domain within HBsAg can be overcome, and MVB-dependent release of HBV is enabled [51,200].

Moreover, 2-adaptin, which is known to participate in the formation of intracellular transport vesicles, was described as a factor for the ESCRT/MVB-dependent release of HBV. The interaction of 2-adaptin with HBsAg exclusively occurs with the cytosolic PreS topology of LHBs. In addition to the interaction with LHBs, 2-adaptin was found to recognize the HBV core protein. This depends on ubiquitinated Nedd4, which binds to the PPAY motif of the core protein. Moreover, the ESCRT-III complex-forming CHMP proteins and the Vps4 ATPases are involved in the release of HBV virions [202]. In accordance with this, it was found that a fraction of HBV is released in the form of exosomes. Interestingly, these exosomes are characterized by the presence of LHBs on their surface, enabling the interaction with NTCP as a bona fide receptor for HBV [203]. In this context, it is interesting that HDV affects autophagy processes. Autophagy seems to be relevant for HDV replication and release [188]. The establishment of a stable cell line persistently replicating and releasing HDV can be a helpful tool for deeper analyses of HDV release [204]. 

Filaments are characterized by a significantly higher fraction of LHBs as compared to spheres. Differing between the genotypes, the ratio of LHBs, MHBs and SHBs is about 1:1:4 [50]. Based on the observation that the MVB-dependent release of HBV depends on the interaction of α-taxilin with LHBs, it was investigated if the presence of a significant fraction of LHBs in filaments leads to an MVB-dependent release of filaments in contrast to the release of spheres by the constitutive secretory pathway. Indeed, it was revealed that filaments like virions are released dependent on ESCRT/MVB in contrast to the secretion via the constitutive secretory pathway of spheres [189,193].

There is not too much known about the release of HDV. Due to the presence of significant amounts of LHBs in the surface protein of HDV, it can be assumed that HDV could be released like HBV virions or filaments dependent on ESCRT/MVB, but there are no clear experimental data addressing this point. It cannot be concluded that the HDV release follows the same route as observed for HBV. For example, there are differences with respect to the impact of N-gylcosylation on the morphogenesis and release of both viruses [19,185,186,187,188]. HBV virion release requires N-linked glycosylation at N146 in the S domain. However, the removal of N-glycosylation sites on the S-, M- and LHBs proteins does not affect the release of but has a partial inhibitory effect on the formation of HDV virions and does not affect their infectivity [205,206,207]. In light of this, it remains unclear if the release of HDV follows the release pathway described for HBV virions and filaments or the route identified for spheres. In accordance with this, there are no data describing the interference of HDV with the ESCRT machinery or the potential exosomal release of HDV and the potential impact of the HBV genotypes on the release pathway for HDV.

The farnesylation of L-HDAg as an essential step in virion assembly of HDV can be prevented by lonafarnib [208]. The CXXX motif at the C-terminal end of L-HDAg serves as a recognition site for the addition of a farnesyl residue to C211 [208]. Monotherapy with lonafarnib led to decreasing levels of HDV RNA, while increasingly higher doses were associated with adverse effects [209,210]. Later, a combination with ritonavir, a cytochrome P450-3A4 inhibitor, enabled higher post-absorbance drug rates while utilizing lower lonafarnib doses [211]. Further studies evaluating combination therapy with lonafarnib + ritonavir and pegylated interferon α were well tolerated and effective at low doses, paving the way for the phase 3 registration study [212]. The phase 3 D-LIVR study found that combination therapy was well tolerated, liver histology significantly improved, and the amount of primary endpoints of at least a 2-log decrease in HDV RNA was increasingly met [213]. As such, the blockage of farnesylation-mediated targeting of L-HDAg towards HBsAg is a promising treatment option for chronic HDV.

## 9. Impact of HDV on HBV Expression

Since HDV is a defective virus, HDV relies on the surface proteins of HBV. Accordingly, both viruses compete for HBsAg. In 1991, in vitro experiments of Wu et al. using transiently transfected hepatoma cells evidenced a competitive interaction of HDV and HBV for the first time. In particular, they observed reduced HBV mRNA levels in the HDV/HBV cotransfected compared to HBV mono-transfected cells. This observation already indicated a dominance of HDV by active suppression of HBV replication [214]. Clinically, decreased HBV DNA viral loads in patient sera as well as decreased intrahepatic HBV replicative intermediates were observed upon HDV coinfection compared to HBV monoinfection in many cross-sectional studies, reflecting the previous in vitro observations [215,216,217]. However, a longitudinal study by Schaper et al. revealed a predominance of HBV over HDV in 10 out of 33 coinfected patients, indicating a more dynamic and fluctuating relation between HBV and HDV in vivo [218]. To manifest the predominance within infected host cells, both viruses interact with host cell factors to interfere with the replication of each other (Figure 4). For instance, follow-up studies of Williams et al. revealed that the suppressive effect of HDV on HBV that was initially observed by Wu et al. is attributed to a direct trans-repression of the HBV enhancer elements by HDAg [219]. The enhancer elements modulate HBV RNA transcription via a cooperative binding of the host factors signal transducer and activator of transcription 3 (STAT-3) and hepatocyte nuclear factor (HNF-3) [220,221]. Using reporter constructs of the HBV enhancer elements, Williams et al. evidenced strong repression of the HBV enhancers by both S- and L-HDAg up to 80%. Interestingly, reporter constructs bearing only the HNF-3/STAT-3 binding site within the HBV enhancers revealed opposing effects of S-HDAg and L-HDAg on STAT-3/HNF-3 binding. L-HDAg but not S-HDAg activated STAT-3/HNF-3 binding to the HBV enhancer. Thus, they hypothesized that S-HDAg might repress HBV RNA transcription by interfering with the STAT-3/HNF-3 binding to the HBV enhancers, whereas L-HDAg, which is produced at a later stage of HDV replication, might activate STAT-3 binding to the HBV enhancer to promote HBV envelope protein synthesis, which is necessary for HDV virion assembly, but maintain suppression of global HBV replication by acting on the 5’ modulatory elements of the HBV enhancer [219]. In line with that, studies by Freitas et al. showed persistent HDV virion assembly and egress in HBV-DNA integrated cell lines with no sign of HBV replication, indicating the replication competence of HDV independent of an active HBV replication by using only HBV-integrated-cell-derived envelope proteins [185]. Likewise, in a thorough in vitro analysis, Wang et al. compared the entry, replication, assembly and release competence of HDV in hepatocytes, which were transfected with cDNA encoding only for HBV envelope proteins of different genotypes [19]. 

At the beginning of chronic HBV infection, HBeAg is largely expressed and secreted into the serum. Spontaneous mutations in the HBV genome particularly in the basal core promoter (BCP) or in the precore region (PC) lead to reduced or a complete loss of HBeAg expression. The loss of HBeAg and the formation of antibodies against HBeAg is termed HBeAg seroconversion [222]. This HBeAg seroconversion accompanied by generally lower HBV DNA levels is associated with slower disease progression and the possibility of spontaneous HBsAg seroconversion, conferring an excellent prognosis [223,224]. In vitro studies as well as clinical studies indicated enhanced replication of HBV strains bearing BCP mutations compared to wild-type strains [225,226,227,228,229,230]. It was hypothesized that the PC mutation might stabilize the HBV RNA encapsidation signal (ε structure) and thereby enhance HBV replication [231,232,233]. Clinically, most of the HDV-infected patients investigated so far were HBeAg-negative, indicating beneficial conditions for HDV replication in HBeAg seroconverted patients. Interestingly, HBeAg-negative patients coinfected with HDV showed strong HBV suppression reflected by significantly reduced intrahepatic and serum HBV levels [234]. This indicates that HDV might have evolved mechanisms to overcome the stabilizing effect of the PC mutation on the ε structure. If host factors such as ISG20 are involved, this mechanism needs to be studied more extensively. Moreover, studies by Kuhnhenn et al. also revealed a perinuclear accumulation of the HBV surface proteins derived from HBeAg-negative HBV patient isolates [233]. However, little is known about the molecular mechanism or the host factors involved in this perinuclear retention of HBsAg. Studies by Chisari et al. indicated that high expression of LHBs might cause perinuclear retention by blocking HBsAg trafficking to the Golgi apparatus [235]. Whether such perinuclear retention also appears in HDV-coinfected cells and whether it is advantageous for HDV assembly or how HDV even overcomes the HBsAg retention in HBeAg-negative patients are interesting questions for prospective studies. 

Furthermore, HBsAg bears a highly conserved N-glycosylation site at position N146. Substitution of this glycosylation site is detrimental to HBV virion release [236]. It was speculated that the depletion of the N-glycosylation site might alter the topology of HBsAg and thereby hinder the binding of HBsAg to the HBV nucleocapsid. Interestingly, the substitution of these glycosylation sites was shown to be tolerated by HDV, since HDV release, assembly and even infectivity were only slightly reduced upon N146Q mutation [205,237]. Hence, modulating the glycosylation of HBsAg by interacting with N-glycosylating enzymes could possibly be another mechanism by which HDV utilizes host factors to suppress HBV replication. Thus, investigating the HBsAg glycosylation pattern of HBV and HDV envelope might be instructive to understanding the interplay of HDV and HBV more in depth. 

However, as discussed in this section, the interplay between HDV and HBV is highly dynamic involving many host factors. Thus, whether active HBV replication gives rise to accessory factors promoting HDV replication needs to be further elucidated. 

## 10. Increased Quasispecies Formation of HBV by Coinfection with HDV

Studies conducted by Godoy et al. analyzing HBV quasispecies indicated a higher HBV quasispecies complexity in chronic HDV-infected patients compared to HBV monoinfected patients [238]. Yamaguchi et al. revealed that HDAg functionally interacts with the RNA Poll II clamp. They further evidenced that HDAg affects the selection of the incorporated nucleotides. Considering previous studies, which indicated that HDAg accelerates the forward translocation of Pol II, Yamaguchi et al. postulated that S-HDAg might increase the transcriptional fidelity of host cell RNA pol II by loosening its clamp [239,240]. Accordingly, Godoy et al. hypothesized that this loosening of the RNA pol II clamp might increase the overall transcriptional error rate, leading to more mutations, for instance, in the 3.5 kb pgRNA and ultimately to more HBV quasispecies [238]. However, such a mechanism would also lead to mutations in host cell mRNAs. Whether such transient mutations are severe for cellular homeostasis and therefore are indeed a pathogenic mechanism of HDV needs to be investigated more extensively. 

Interestingly, a study by Sajjad et al. revealed unique mutations in the HBsAg gene, which were only observed in HDV/HBV-coinfected patients. For instance, the tryptophan at position 196 of HBsAg was mutated to leucine in almost all HDV/HBV-coinfected patients, although none of the HBV-monoinfected patients carried this mutation [241]. The tryptophan at position 196 was previously shown to be essential for HBsAg and L-HDAg interaction. Mutation of amino acid 196 was shown to inhibit HDV release [242]. Thus, these observations indicate that HBV counteracts the suppressive effect of HDV by selectively amplifying mutations that are essential for HDV replication. How HBV facilitates this selective mutation and which cellular host factors are involved remain elusive. Nevertheless, this counteraction of HBV by selective mutation might explain the fluctuation in the predominance of both viruses observed in the longitudinal study of Sharper et al. [218]. 

## 11. Impact of HDV on Mechanisms of the Innate Immunity

William et al. also showed the transactivation of the interferon-inducible GTP-binding protein MxA by L-HDAg. MxA has intrinsic antiviral properties and was evidenced to be a potent inhibitor of HBV replication [243,244]. In 2012, Li et al. postulated a mechanism in which MxA hinders HBV capsid assembly and consequently HBV virion assembly by binding to the HBV core protein and trapping it in perinuclear compartments [245]. The activation of the MxA gene by L-HDAg thus facilitates the interference of HBV replication without affecting the production of the HBV surface proteins [219]. 

Moreover, a very recent study by Lucifora et al. characterized additional interferon-stimulated genes (ISGs) that were upregulated upon HDV infection in either differentiated HepaRG cells or primary human hepatocytes. According to their study, ISG upregulation is induced by the amplification of the genomic and antigenomic HDV RNA rather than the expression of the HDAg proteins [246]. However, this is in contrast to the abovementioned L-HDAg-mediated MxA activation observed by Williams et al. [219]. Lucifora et al. identified 73 ISGs that were significantly upregulated during HDV infection; amongst others were the cellular host factors Tripartite motif-containing 22 (TRIM22), DExD/H-Box Helicase 60 (DDX60) and interferon-stimulated exoribonuclease gene of 20 kDa (ISG20), which were all suggested to have anti-HBV activity [246]. More precisely, TRIM22 was speculated to inhibit HBV core promoter activity by forming a nuclear scaffold that interacts with the HBV transcription factors and thereby acting as a transcriptional repressor [247]. DDX60, a cytoplasmic RNA helicase, promotes the degradation of HBV RNAs in the cytoplasm [248]. ISG20 specifically degrades HBV RNAs by recognizing the N6-methyladenosine modification on the epsilon stem-loop structure (ε), which is present in all HBV RNAs. Since ε is also the pgRNA packing signal as well as the priming site of reverse transcription, ISG20 inhibits HBV genome replication by preventing the binding of the viral polymerase to ε of HBV pgRNA [249,250]. According to Lucifora et al., mutating the recognition motif of ISG20 in ε structure did not affect HBV suppression. Thus, one can speculate that either ISG20 is not involved in HDV-induced HBV suppression or that the effect can be compensated by other HDV interference mechanisms. Moreover, inhibiting the interferon pathway only partially reversed the suppressive effect of HDV on HBV, emphasizing the involvement of other interferon-response-independent host factors or a direct interaction of the viral components in HBV suppression [246]. 

Studies by Belloni et al. have described that IFNα treatment induces hypoacetylation of HBV cccDNA-bound histones, leading to recruitment of the transcriptional repressor yin yang 1 (YY1) and polycomb protein enhancer of zeste homolog 2 (Ezh2), which in turn downregulate HBV transcription epigenetically [251]. Previous studies already revealed the capability of HDV to epigenetically control certain genes by altering histone acetylation [252]. We have also already discussed the activation of many ISGs by HDV. Thus, one can speculate that HDV might also exert an epigenetic control on HBV by recruiting YY1 and Ezh2 to its cccDNA. 

Taken together, HDV and HBV both compete for predominance in the infected host cell. HDV has developed various mechanisms involving many cellular host factors to suppress active HBV replication but still maintain HBV envelope protein synthesis, which is nonetheless essential for HDV replication. Although several studies have focused on the dynamic interplay between HDV and HBV, more investigation is needed to elucidate key host factors.

## 12. Cellular Factors Involved in HDV Pathogenesis

As previously discussed, chronic HDV infection induces the most severe form of liver disease amongst all hepatitis viruses. HDV/HBV coinfection leads to accelerated liver cirrhosis and increased risk for HCC compared to HBV monoinfection. This high pathogenicity of HDV can partially be attributed to the capability of HDV to dysregulate cellular functions by regulating a vast amount of cellular host factors (Figure 5). This is evidenced by the proteomic analysis of Mendes et al. showing that HDV replication altered the expression of 89 of a total of 3000 investigated proteins [253] and by the recent transcriptomic analysis of Lucifora et al. revealing 73 genes that were upregulated by HDV replication [246]. 

In line with that, studies by Goto et al. revealed a synergistic activation of the serum response element (SRE)-dependent pathway by HBV and HDV. In particular, HBV HBx activated the transcription of the transcription factor Elk1, whereas L-HDAg activated the transcription of the serum response factor (SFR), which in turn led to synergistic activation of SRE-regulated host factors. Accordingly, one of the SRE-regulated host factors, c-Fos, was proven to be activated by this means [254]. c-Fos is a proto-oncogene that binds to another oncogenic transcription factor, c-Jun, to form a heterodimer. The resulting heterodimer termed activator protein 1 (AP-1) is a transcription factor, which controls many cellular processes including cell proliferation, cell differentiation and apoptosis [255]. Accordingly, aberrant expression of c-Fos induced by L-HDAg and HBx alters AP-1-dependent gene expression [254]. Likewise, Choi et al. postulated another mechanism by which HDV and HBV synergistically activate AP-1-dependent gene expression. They evidenced that L-HDAg can bind to the basic leucine zipper domain of c-Jun and thereby facilitate the activation of the AP-1 signaling cascade [256]. Previous studies indicated an activation of the c-Jun signaling cascade by HBV HBx [257,258]. In line with that, Choi et al. revealed a synergistic activation of the c-Jun-mediated signal cascade upon L-HDAg and HBx coexpression, indicating that HBV and HDV evolved various mechanisms to alter AP-1-regulated gene expression [256]. Activation of AP-1 was shown to be a frequent and early event in HCC probably due to an involvement of AP-1 in hepatocyte transformation [259,260,261].

The same study also revealed a synergistic activation of the Transforming growth factor-β (TGF-β) signaling cascade by HBV HBx and HDV L-HDAg [256]. TGF-β is a fibrogenic cytokine that plays a pivotal role in liver regeneration. Therefore, TGF-β has high relevance for fibrotic and cirrhotic processes [262]. For instance, TGF-β induces the expression of plasmogen-activator inhibitor type-1 (PAI-1), which in term inhibits fibronylisis. Thus, the overexpression of PAI-1 is associated with hepatic fibrosis and cirrhosis [263,264,265]. According to Choi et al., the protein expression of PAI-1 in TGF-β-stimulated Huh7 cells was markedly increased upon expression of L-HDAg, revealing yet another mechanism for HDV pathogenicity. Since S-HDAg as well as a prenylation-deficient L-HDAg mutant did not activate either the AP-1 or the TGF-β signaling cascade, Choi et al. postulated that the mechanism of activation involves the prenylation of L-HDAg [256]. A recent study by Liang et al. confirmed this observation. They reported that L-HDAg interacts with SMAD3, a transducer of the TGF-β signaling cascade, and thereby activates the expression of the transcription factor Twist. Interestingly, inhibition of L-HDAg prenylation by statins also inhibited Twist promoter activation [266]. Considering that TGF-β-induced PAI-1 activation is also mediated by SMAD3 binding to the PAI-1 promoter, these results emphasize the pivotal role of L-HDAg prenylation in SMAD3-mediated activation of the TGF-β signaling cascade [263]. Moreover, since Twist is involved in the epithelial–mesenchymal transition process leading to fibrosis and HCC, the activation of Twist expression by L-HDAg reflects yet another mechanism of HDV-induced pathogenesis [266,267,268]. 

Another mechanism by which HDV might promote HCC is by inhibiting clathrin-mediated protein transport. L-HDAg is an adaptor-like protein that directly interacts with the clathrin heavy chain (CHC) at the trans-Golgi network to facilitate HDV virion assembly. However, the binding of L-HDAg to CHC interferes with the binding of other cellular adaptor proteins to CHC, leading to an inhibition of the clathrin-dependent transport of host proteins [269,270,271,272]. For instance, Huang et al. described that internalization of the major iron transport protein transferrin is reduced by L-HDAg expression, which results in the disruption of iron homeostasis. Likewise, transport to lysosomes and the degradation of ligand-bound epidermal growth factor (EGFR) is impaired upon L-HDAg expression. Defects in the degradation of ligand-bound EGFR could lead to the constitutive activation of cell proliferation and ultimately to oncogenesis [272,273,274,275]. Interestingly, previous studies indicated that HBV preferentially replicates in quiescent cells, whereas in proliferating cells, HBV replication is reduced. Studies by Friedrich et al. even described cell-cycle-inhibitory functions of HBV to promote its replication. Thus, the activation of the pro-proliferative pathway by HDV might reflect yet another mechanism of HDV-induced HBV suppression. 

Moreover, studies by Williams et al. revealed that HDAg expression enhances the production of reactive oxygen species (ROS) by activating the expression of NADPH oxidase 4 (Nox4). Nox4 in turn produces H2O2, constitutively leading to oxidative stress. Since the Nox4 gene is also controlled by TGF-β, Williams et al. suggested that L-HDAg might activate Nox4 expression through the TGF-β signaling cascade as previously described. However, L-HDAg expression in TGF-β-treated hepatocytes did not significantly enhance NOx4 expression, indicating TGF-β-independent pathways involved in L-HDAg-induced Nox4 activation [276]. Recently, Smirnova et al. described that L-HDAg also activates the expression of NADPH oxidase 1, cytochrome P450 2E1 and ER oxidoreductin 1 α [277]. All of these enzymes have previously been shown to induce oxidative stress upon consecutive expression [278,279]. Moreover, studies by Chen et al. suggested another mechanism by which HDV induces oxidative stress. According to their study, S-HDAg directly and specifically binds to the mRNA of the glutathione S-transferase P1 (GSTP1) gene and thereby downregulates its expression leading to ROS accumulation and increased apoptosis [280]. GSTP1 is considered a phase II detoxification enzyme, which plays a crucial role in reducing ROS and protecting cells against oxidative DNA damage. Several clinical studies reported that GSTP1 is significantly downregulated in HCC patients, and downregulation correlates with high oxidative stress [281]. Oxidative stress is a key factor in carcinogenesis, since it drives genomic DNA damage and genetic instability leading to mutagenesis, which is a central precipitant for tumorigenesis [282]. Additionally, several in vivo studies evidenced that elevated ROS levels inhibit liver regeneration which in turn promotes liver damage [283]. Interestingly, HBV has opposing effects on oxidative stress compared to HDV. In vitro analysis of HBV-replicating cells and of human liver samples derived from patients suffering from chronic HBV infection indicated activation of the Nrf2/antioxidant response element (ARE) pathway by the HBV proteins HBx and LHBs [284]. The activation of Nrf2/ARE in turn induces the expression of cytoprotective genes, leading to the inactivation of ROS [285]. It is speculated that this mechanism represents a viral strategy to escape the elimination of infected cells, which are recognized by the host immune system due to their elevated ROS levels. Suppression of HBV by HDV might therefore also prevent activation of the Nrf2/ARE pathway, leading to even higher oxidative stress in HDV/HBV coinfection compared to HBV monoinfection and consequently to more severe pathogenesis. 

In 2017, Suárez-Amarám et al. facilitated the development of an HDV mouse model using adeno-associated viral vectors [286]. Using this model, Usai et al. further aimed to determine essential viral and host factors associated with HDV-induced liver injury. They observed hepatic injury characterized by necrosis and apoptosis in HDV/HBV-coinfected mice. Interestingly, the hepatic injury was not induced by the host immune response since inhibition or depletion of the immune response did not prevent hepatic injury in the infected mice. Therefore, they concluded that hepatic injury was presumably induced by a direct cytotoxic effect of S- and L-HDAg. Transcriptomic analysis of the injured liver revealed strong activation of the tumor necrosis factor-α (TNF-α) signaling pathways. Accordingly, inhibition of TNF-α resulted in amelioration of the hepatic injury in the infected mice [287]. TNF-α is an inflammatory cytokine that activates the expression of hundreds of proteins such as the Nuclear factor-κB (NF-κB). NF-κB is considered a master regulator of inflammation and apoptosis. It is activated in virtually any chronic liver injury. Moreover, NF-κB activation is strongly associated with HCC progression [288]. Interestingly, previous studies by Park et al. in 2009 already showed an enhancement of TNF-α-induced NF-κB activation by L-HDAg, indicating a key role of NF-κB in HDV pathogenesis [289]. Nevertheless, whether consecutive NF-κB activation is indeed causative for hepatic injury and HCC development in HDV infection needs to be investigated more extensively.

Furthermore, studies by Liao et al. revealed upregulation of the clusterin gene expression by HDAg [252]. Clusterin is a molecular chaperone, which is involved in the clearance of cell debris and apoptosis. The overexpression of clusterin is reported to play a role in HCC metastasis [290]. Interestingly, studies by Liao et al. indicated that the upregulation of clusterin was mediated by histone hyperacetylation. Therefore, they speculated an interaction of HDAg with either histone acetyltransferases or deacetylases in modulating histone acetylation [252]. Nevertheless, such an interaction needs to be evidenced, and the epigenetic upregulation of other proto-oncogenes through histone acetylation by HDV needs to be further elucidated. 

Another mechanism of epigenetic regulation is DNA methylation. Alterations in DNA methylation, for instance, hypermethylation of tumor suppressor genes, is strongly associated with carcinogenesis [291]. In 2013, Benegiamo et al. showed that HDV transactivates the overexpression of DNA-methyltransferase 3b (DNMT3b) through STAT3 activation. Moreover, they showed that the overexpression of DNMT3b led to hypermethylation and consequently to the downregulation of the transcription factor E2F1 [292]. E2F1 is a crucial mediator of cell cycle progression and apoptosis [293,294]. Interestingly, Benegiamo et al. observed a two-fold increase in G2/M cell cycle arrest in HDV-replicating Huh7 cells probably linked to E2F1 silencing. Since E2F1 silencing can also lead to disrupted DNA repair and inhibition of apoptosis by escape mutations, the epigenetic control of E2F1 might reflect another mechanism of HDV-induced oncogenesis [292].

In 2016, Huang et al. described that L-HDAg directly interacts via its C-terminal domain with the nuclear RNA export factor 1 (NXF1). According to their study, this interaction is crucial for HDV virion assembly and release. Since the inhibition of NXF1 results in nuclear accumulation of poly (A)+ RNAs, the authors speculated that potential inhibitory effects of L-HDAg on the biological function of NXF1 might result in impaired liver regeneration in the long term [117]. However, potential inhibitory effects on NXF1 and a consequent impact on hepatic disease need to be elucidated [295].

Recent studies of Khabir moreover reported that HDV alters the autophagy process to promote its genome replication. In particular, they showed that both HDAg isoforms can efficiently promote autophagosome accumulation by blocking autophagic flux. Further, they evidenced that the autophagy related 5 (ATG5) protein is the key proviral factor that assists HDV genome replication in a non-canonical manner. Considering that HDV genome replication occurs in the nucleus, relocalization of the autophagy proteins to the nucleus during HDV genome replication might impact the autophagy process. Khabir et al. evidenced autophagasome accumulation by accumulation of the ubiquitin-binding protein p62. Accumulation of p62 precedes HCC development and thus is considered a biomarker for chronic liver disease and HCC progression [188,296,297]. 

In 2018, a very thorough analysis by Diaz et al. aimed to decipher the unique molecular signature of HDV-associated HCC. To this end, transcriptomic profiling of either HDV, HBV only or HCV-associated HCC liver tissues was conducted. This thorough analysis revealed 20 top-scored signaling pathways that were dysregulated exclusively in HDV-HCC. For instance, genes associated with the STAT3 pathway were dysregulated in HDV-HCC, which is in line with studies by William et al. reporting STAT3 activation by L-HDAg [276,298]. Interestingly, previous studies by Pugnale et al. indicated an inhibition of the IFN-induced JAK-STAT signaling pathway by HDV through impairment of the phosphorylation of STAT1 and STAT2 [299]. This emphasizes the capability of HDV to fine-modulate the activity of the STAT protein isoforms. Furthermore, although most of the dysregulated genes in HDV-HCC were downregulated, Diaz et al. identified six signaling pathways that were predominantly and exclusively upregulated in HDV-HCC (sonic hedgehog signaling, GADD45 signaling, DNA damage-induced 14-3-3σ signaling, cyclins and cell cycle regulation signaling, cell cycle: G2–M DNA damage checkpoint regulation and hereditary breast cancer signaling). Interestingly, these pathways are involved in DNA replication, damage and repair as well as genome instability. Since these genes were unaffected in the HBV-HCC specimens, the authors suggested that HDV induces host genome instability and that this genome instability is an important and moreover unique pathogenic trait of HDV [298].

Conclusively, these studies emphasize the high oncogenic potential of HDV. Whether HDV is an oncogenic virus is still a matter of debate due to its dependency on a helper virus to complete its replication cycle. However, considering the vast amount of pathogenic mechanisms that are unique to HDV and even more severe than those of HBV, it is more or less evident that HDV is a potent oncogenic virus. Nevertheless, more studies need to be conducted to elucidate all of the many mechanisms underlying HDV pathogenesis and to decipher more key host factors, which could be targeted to treat hepatitis D.

## Figures and Tables

**Figure 1 viruses-15-01687-f001:**
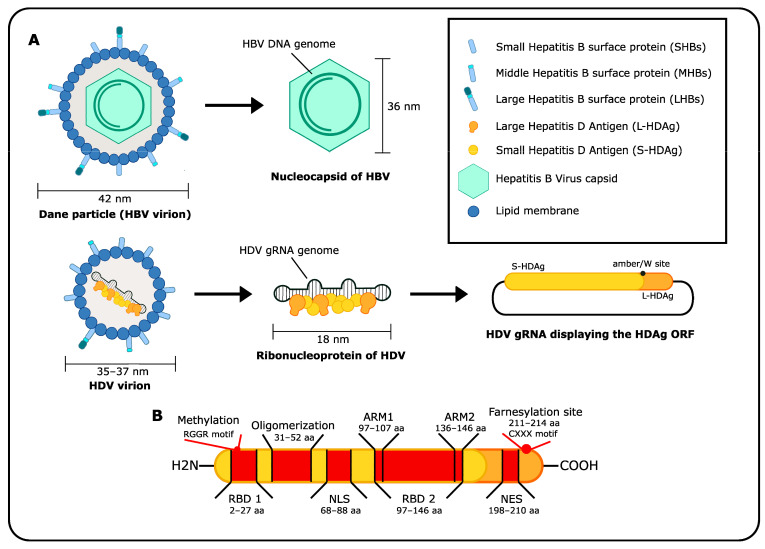
Illustration of the viral particles of Hepatitis D and Hepatitis B virus. (**A**) Sizes of the respective particles are indicated by the bars shown. (**B**) S- and L-HDAg with their respective functional domains indicated by red boxes. Sites for methylation or farnesylation are indicated by red circles. RBD: RNA-binding domain; NLS: Nuclear localization sequence; NES: Nuclear export sequence; ARM: Arginine-rich motif.

**Figure 2 viruses-15-01687-f002:**
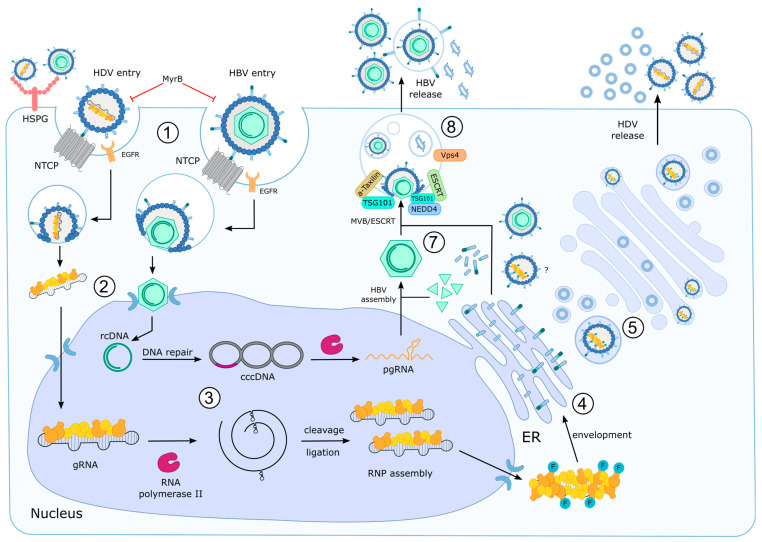
Interactions with the host during entry and egress. (1) Initially, low-affinity interaction of S-HBs with HSPG is followed by the interaction of the preS1 domain and NTCP. Subsequent internalization of bound virus and NTCP with EGFR as cofactor occurs most likely in a clathrin-mediated manner. (2) The viral nucleocapsid is released from the endosome and transported into the nucleus where viral replication takes place (3) (see Section 7). Upon assembly of the viral ribonucleoprotein (RNP) around the gRNA of HDV, the RNP is exported from the nucleus and enveloped by budding into the ER lumen (4). The enveloped RNP is subsequently believed to be released through the secretory pathway, following the spherical SVPs of HBV (5). Pregenomic RNA of HBV is encapsidated in the cytosol by the core protein and subsequently enveloped by either budding into the ER lumen or directed towards MVBs through the interaction with a-taxilin and tsg101 (7). Similarly, nucleocapsids of HBV enveloped by budding into the ER lumen can be transported to MVBs and ultimately be released inside of exosomes (8). MVB-dependent or exosomal release of HDV RNPs is a possibility but remains to be investigated (7).

**Figure 3 viruses-15-01687-f003:**
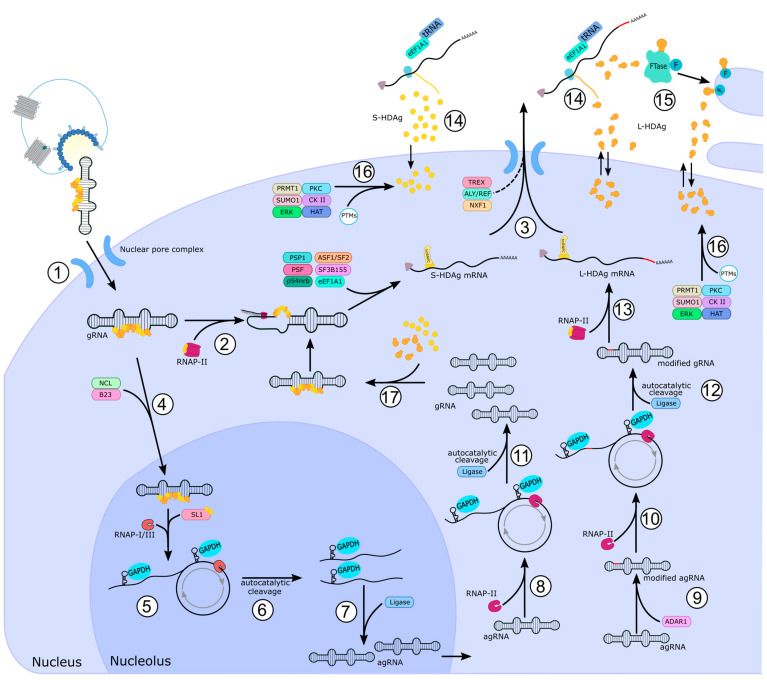
Interaction of HDV with the host during replication. Upon release of the viral RNP from the endosome, the RNP complex is transported into the nucleus through the nuclear pore complex (NPC) via interaction with karyopherin 2a (1). In the nucleus, S-HDAg recruits and modulates RNAP-II to facilitate mRNA transcription. Transcription is promoted by various host cell factors (2). The replication intermediate of HDV is generated upon shuttling of gRNA into the nucleolus, possibly via interaction with B23 and NCL (4), where RNAP-I is recruited and carries out the rolling circle replication with help of GAPDH (5). Next, auto-catalytic cleavage by the HDV ribozyme (6) and host-provided ligase generates quasi-double-stranded agRNA molecules (7), which either serve as template for the production of gRNA (8, 11) or as target for adenosine deaminase acting on RNA (ADAR1) (9) to generate the template for the L-HDAg mRNA (10, 12, 13). Synthesized S- and L-HDAg mRNA are likely exported through interaction with Aly/REF, the nuclear RNA export factor (NXF1) and other components of the Transcription export (TREX) complex (3). Translation of S- and L-HDAg in the cytosol might be assisted by eukaryotic translation elongation factor 1 alpha 1 (eEF1A1) and recruitment of tRNA (14). Subsequent re-import of S- and L-HDAg into the nucleus and phosphorylation facilitated by different kinases as well as other post-translational modifications (PTMs) are responsible for the regulation and different functions of the HDAg isoforms (16). Farnesylation of L-HDAg is carried out by farnesyltransferase (FTase) and is essential for the interaction with HBsAg and subsequent release (15). S- and L-HDAg associate with the gRNA to form novel HDV RNPs (17).

**Figure 4 viruses-15-01687-f004:**
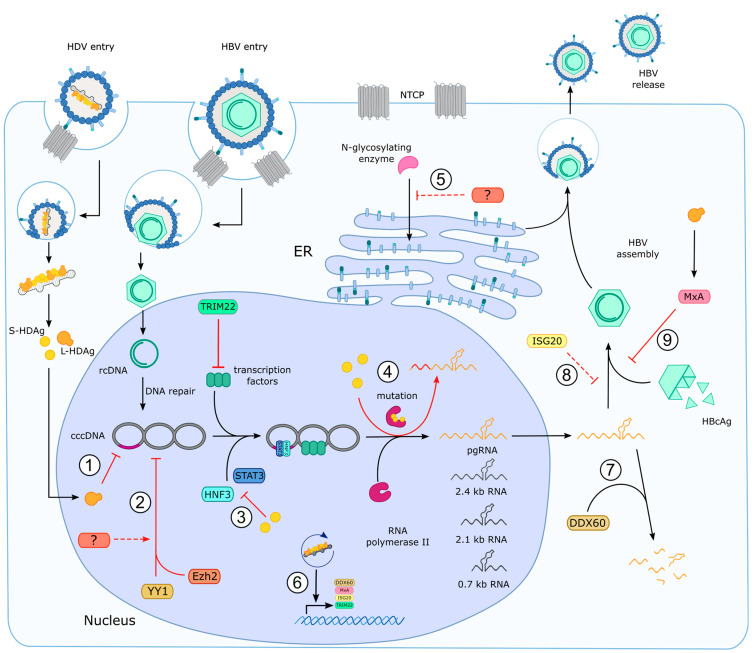
Schematic representation of HDV interference mechanisms on HBV replication. HDV suppresses HBV RNA transcription likely through L-HDAg acting on HBV enhancer elements (1) and through inhibition of binding of the transcription factors STAT3/ HNF3 to HBV enhancer elements by S-HDAg (3). Recruitment of HBV cccDNA transcriptional repressors YY1 and Ezh2 by epigenetic modifications induced by HDV can be speculated (2). Interaction of S-HDAg with RNA polymerase II (RNAP II) increases error rate of RNA pol II and subsequently leads to mutations in HBV pregenomic RNA (pgRNA) (4). Interference of HBV HBsAg glycosylation, which is detrimental for HBV replication, by interaction of HDV with N-glycosylating enzymes can be speculated (5). HDV replication activates expression of the interferon-stimulated genes MxA, DDX60, ISG20 and TRIM22 (6). DDX60 degrades cytoplasmic HBV RNAs (7). ISG20 inhibits packaging of HBV pgRNA into HBV capsids (8). MxA, which can also be activated by L-HDAg only, inhibits HBV capsid assembly (9).

**Figure 5 viruses-15-01687-f005:**
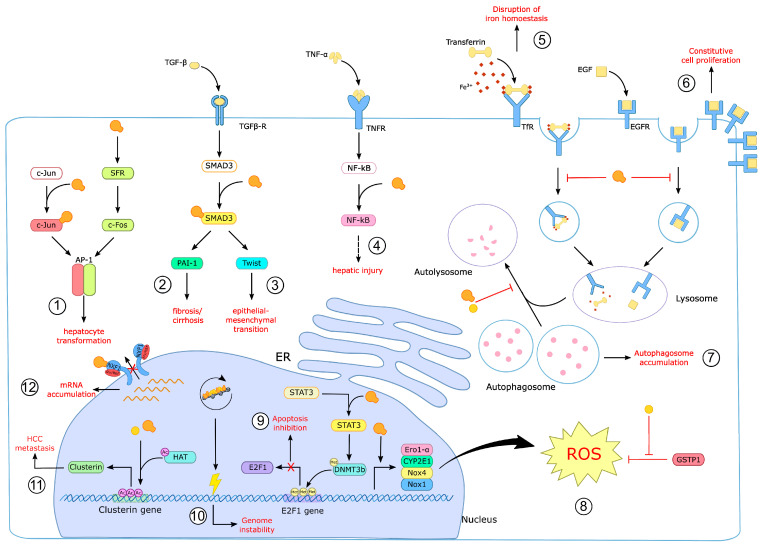
Schematic representation of cellular host factors involved in HDV pathogenesis. L-HDAg activates the proto-oncogene c-Jun by direct interaction and the proto-oncogene c-Fos via activating the serum response factor (SFR). C-Jun and c-Fos dimerize to form the transcription factor AP-1, which in turn is supposed to be involved in hepatocyte transformation (1). L-HDAg promotes the TGF-β signaling cascade by interacting with the TGF-β signal transducer SMAD3. Expression of the plasmogen-activator inhibitor type-1 (PAI-1), which promotes fibrosis and cirrhosis (2), and the transcription factor Twist, which might induce epithelial–mesenchymal transition leading to hepatocellular carcinoma (HCC) (3), are activated by this means. L-HDAg also promotes the TNFα signaling cascade, in particular expression of the master regulator of apoptosis NF-κB, which might induce hepatic injury (4). L-HDAg inhibits clathrin-mediated protein transport. For instance, internalization of the major iron transport protein, transferrin (5) and ligand-bound epidermal growth factor receptor (6) is inhibited by L-HDAg, leading to disrupted iron homeostasis (5) and constitutive cell proliferation (6). Both S- and L-HDAg inhibit autophagic flux, leading to autophagosome accumulation (7). L-HDAg activates the cellular host factors Ero1-α, CYP2E1, Nox4 and Nox1, which in turn increase cellular ROS level, leading to oxidative stress. Moreover, S-HDAg inhibits GSTP1 expression, which acts as a detoxifying enzyme, reducing cellular ROS level (8). L-HDAg transactivates DNA-methyltransferase 3b (DNMT3b) via activation of STAT3. DNMT3B in turn hypermethylates the E2F1 gene, which leads to downregulation of E2F1, a crucial mediator of apoptosis, and consequently to apoptosis inhibition (9). Further, S- and L-HDAg activate expression of clusterin, a molecular chaperone that is strongly associated with HCC metastasis upon overexpression. Clusterin activation is probably mediated by interaction of S- and L-HDAg with histone acetyltransferase, which in turn hyperacetylate the clusterin gene (11). L-HDAg interacts with the nuclear RNA export factor 1 (NXF1). Impairment of NXF1 function by the L-HDAg interaction leading to accumulation of cellular mRNAs can be speculated (12). Moreover, HDV replication in general activates signaling cascades, which are strongly associated with genome instability (10).

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
