# Peer review of "Cellular Factors Involved in the Hepatitis D Virus Life Cycle"

_viruses, 2023, doi:10.3390/v15081687_

Round 1

Reviewer 1 Report

The gist of this manuscript is to demonstrate the possible roles of cellular factors in the biology of hepatitis delta virus (HDV), including  morphogenesis, virion structure, viral replication and viral pathogenesis and so on. This is a very ambitious undertaking because the factors involved in HDV (and part of HBV) have not been fully clarified.  The approach of the current manuscript is to compare the functional roles and biological mechanism of the various cellular factors in the biology of these two viruses.  The problem is that the mechanisms of most of these cellular proteins in the biology of these two viruses, e.g. the composition of the RNA replication complex,  have barely begun to be understood.  This is one of the major unsolved issues in HDV.   All of the figures, in particular, Fig. 2 is too speculative and difficult to understand.  It will be more helpful to simplify the figures.

Title: the title of this manuscript reads very  awkward.

Author Response

Point-by-point reply

Reviewer 1:

The gist of this manuscript is to demonstrate the possible roles of cellular factors in the biology of hepatitis delta virus (HDV), including  morphogenesis, virion structure, viral replication and viral pathogenesis and so on. This is a very ambitious undertaking because the factors involved in HDV (and part of HBV) have not been fully clarified.  The approach of the current manuscript is to compare the functional roles and biological mechanism of the various cellular factors in the biology of these two viruses.  The problem is that the mechanisms of most of these cellular proteins in the biology of these two viruses, e.g. the composition of the RNA replication complex,  have barely begun to be understood.  This is one of the major unsolved issues in HDV.   All of the figures, in particular, Fig. 2 is too speculative and difficult to understand.  It will be more helpful to simplify the figures.

We fully agree with the reviewer that there are a lot of open questions  describing the life cycle of HDV. Therefore, this review summarizes on the one hand published data on this field and on the other hand defines some of the open questions. Our intention is to contribute to fostering research on these open questions. As suggested by the reviewer we modified figures, but we would like to emphasize that the content of the figures is described in detail in the figure legends to make it easier to understand.

Title: the title of this manuscript reads very  awkward.

As suggested by the reviewer we modified the title of the revised manuscript.

Reviewer 2 Report

In this manuscript, Thiyagarajah and colleagues reviewed the molecular interactions between HDV and hepatocyte host factors. This is a well-written, highly comprehensive review with extended and clear descriptions of these interactions. I only have minor comments. 

Specific comments:

1. HDV is a peculiar virus highly difficult to target directly in antiviral therapies. In this context, the innovative antiviral strategies are mostly based on host targeting agents (HTA), requesting a comprehensive analysis of virus host interactions. although mentioned in the abstract, these aspects are not developed in the manuscript. A small update in the current trials involving HTA, especially the first release of lonafarnib phase III results (Etzion et al., EASL Congress 2023) and bulevirtide phase III results (Wedemeyer et al., N Engl J Med 2023) would deserve to be mentioned. 

2. In the same vein, the authors do not mention at all the inhibition of FTase and its impact on HDV infection. L-HDAg farnesylation is essential for both HDV morphogenesis and L-HDAg-mediated inhibition of HDV replication. In this context, few sentences on the impact of the FTase inhibitor lonfarnib on the HDV life cycle, leading to both an intracellular accumulation of HDV RNA and an inhibition of viral particle secretion as described by several groups (Bach et al., Antiviral Res 2023; Lempp et al., Nat Commun 2019; Sato et al., J Virol 2004) would deserve to be discussed. Moreover, a recent study highlighted the effect of lonafarnib on viral particle infectivity (Verrier et al., Antiviral Res 2022). 

3. In the second paragraph, it could be interesting to mention that many delta-like viruses have been recently described in animals, not necessarily associated with hepadnaviruses and liver infections (for a review: Netter et al., Front Microbiol 2021)

4. Paragraph n°6: the cryo-EM structure of NTCP has been reported by 3 independent groups and this would deserve to be explicitly mentioned (Park et al., Nature 2022; Goutam et al., Nature 2022; Asami et al., Nature 2022)

Author Response

Point-by point reply

Reviewer: 2

  1. HDV is a peculiar virus highly difficult to target directly in antiviral therapies. In this context, the innovative antiviral strategies are mostly based on host targeting agents (HTA), requesting a comprehensive analysis of virus host interactions. although mentioned in the abstract, these aspects are not developed in the manuscript. A small update in the current trials involving HTA, especially the first release of lonafarnib phase III results (Etzion et al., EASL Congress 2023) and bulevirtide phase III results (Wedemeyer et al., N Engl J Med 2023) would deserve to be mentioned. 
  • According to the reviewer’s suggestion, we added a new paragraph mentioning the phase III study for lonafarnib and bulevirtide (see line 235-240 and lines 621-633)
  1. In the same vein, the authors do not mention at all the inhibition of FTase and its impact on HDV infection. L-HDAg farnesylation is essential for both HDV morphogenesis and L-HDAg-mediated inhibition of HDV replication. In this context, few sentences on the impact of the FTase inhibitor lonfarnib on the HDV life cycle, leading to both an intracellular accumulation of HDV RNA and an inhibition of viral particle secretion as described by several groups (Bach et al., Antiviral Res 2023; Lempp et al., Nat Commun 2019; Sato et al., J Virol 2004) would deserve to be discussed. Moreover, a recent study highlighted the effect of lonafarnib on viral particle infectivity (Verrier et al., Antiviral Res 2022). 
  • We agree with the reviewer and mentioned the impact of FTase inhibition on HDV (see lines 410-417).

  1. In the second paragraph, it could be interesting to mention that many delta-like viruses have been recently described in animals, not necessarily associated with hepadnaviruses and liver infections (for a review: Netter et al., Front Microbiol 2021)
  • As suggested by the reviewer we included this information in the revised version of the manuscript (see lines 68-71).
  1. Paragraph n°6: the cryo-EM structure of NTCP has been reported by 3 independent groups and this would deserve to be explicitly mentioned (Park et al., Nature 2022; Goutam et al., Nature 2022; Asami et al., Nature 2022)
  • In the revised version we included these references as requested by the reviewer (see line 182).

Reviewer 3 Report

This review discussed extensively regarding HDV replication, morphogenesis, and pathogenesis. 

Several suggestions:

1.      It is better to draw a figure to summarize all the information written in lines 82-90, 279-283, 402-406.

2.      Line 106, [the replication of the 3.5 kb] or [the transcription of the 3.5 kb]?

3.      Line 187, [aa 84-77] or [aa 77-84]?

4.      Line 188, please define [RBD]? Receptor binding domain?

5.      Line 211, please write the full name for [EMA] at its first appearance.

6.      Please add references after the sentences at the end of line 287, line 600, line 671. 768, line 877.

7.      Line 339, please change [transcription] to [replication].

8.      Line 343, why there is a [e.g.]?

9.      Line 567, please define [i-PreS].

10.  Why put the information on lines 574-577 after the information on lines 565-574? Are they related?

11.  Line 651, please check the definition of HBeAg seroconversion. [During the natural history of chronic hepatitis B virus (HBV) infection, the loss of serum hepatitis B e antigen (HBeAg) and the development of anti-HBe antibodies (HBeAg seroconversion) mark a transition from the immune-active phase of disease to the inactive carrier state.] from [Hepatol Int. 2009 Sep; 3(3): 425–433.].

12.  Line 662, please define [ε structure] at its first appearance. 

I am not qualified to assess the quality of English. To make the article easier to understand, English editing is suggested.

Author Response

Point by point reply

Reviewer 3:

  1. It is better to draw a figure to summarize all the information written in lines 82-90, 279-283, 402-406.
  • As suggested by the reviewer we included a new figure illustrating the information for a better overview (see new Figure 1, line 94-98, page 3).
  1. Line 106, [the replication of the 3.5 kb] or [the transcription of the 3.5 kb]?
  • We amended this line according to the reviewer’s suggestion (see line 115).
  1. Line 187, [aa 84-77] or [aa 77-84]?
  • As requested by the reviewer, we amended this line according to the reviewer’s suggestion (see line 196).
  1. Line 188, please define [RBD]? Receptor binding domain?
  • We modified the text according to the reviewer’s suggestion and removed the abbreviation of RBD in this context to avoid confusion (see lines 197, 199, 200, 202, 204 and 213).
  1. Line 211, please write the full name for [EMA] at its first appearance.
  • We concur with the reviewer and modified the text accordingly (see line 221).
  1. Please add references after the sentences at the end of line 287, line 600, line 671. 768, line 877.
  • We agree with the reviewers suggestions and added references to the sentences at the end of line 302 (previous 287), line 702 (previous 671 and line 802 (previous 768). At the end of line 620 (previous 600) we did not add references as the release pathway for HDV has not been elucidated and there is no data regarding interference with ESCRT and HDV or regarding exosomal release.
  1. Line 339, please change [transcription] to [replication].
  • As suggested by the reviewer we changed the text accordingly (see line 354).
  1. Line 343, why there is a [e.g.]?
  • Lines 357-358 have been amended to avoid confusion according to the reviewer’s suggestion.
  1. Line 567, please define [i-PreS].
  • i-PreS has been changed to [cytosolic PreS] (line 588).
  1. Why put the information on lines 574-577 after the information on lines 565-574? Are they related?

The formation of ILVs (intraluminal vesicles) and MVBs (multivesicular bodies) is tightly associated with autophagy processes. Therefore, we decided to include this information at this place.

  1. Line 651, please check the definition of HBeAg seroconversion. [During the natural history of chronic hepatitis B virus (HBV) infection, the loss of serum hepatitis B e antigen (HBeAg) and the development of anti-HBe antibodies (HBeAg seroconversion) mark a transition from the immune-active phase of disease to the inactive carrier state.] from [Hepatol Int. 2009 Sep; 3(3): 425–433.].
  • The text was changed according to the reviewer’s suggestion (See lines 684-688).
  1. Line 662, please define [ε structure] at its first appearance.
  • We agree with the reviewer and modified the text accordingly (see line 690).

Round 2

Reviewer 3 Report

The questions I raised previously have been addressed in this revised manuscript.

Author Response

Point-to-point response

Acedemic editor:

  1. ) It is okay to show HDV has less than 1,700 nucleotides in the abstract (line 10). However, it can be more accurate (lines 26-27 and line 55) as shown in literature: 1672-1697 nucleotides.
  • According to the editor´s suggestion, we specified the genome size (see line 26 and 56)
  1. Clarify "both" or "each HDAg ...... respectively" (line 29)
  • According to the editor´s suggestion we mentioned each of the HDAg isoforms separately (see lines 28-31).
  1. Rephrase "a secreted variant of HBcAg, termed HBeAg". HBeAg has a unique fragment at its N-terminus. Present clearly HBcAg, HBcrAg (likely a secreted variant of HBcAg) and HBeAg (lines 109-110)
  • We agree with the editor and rephrased the sentence and moreover explained more in depth the various Hepatitis B core-related antigens (HBcAg, HBeAg, p25, p22cr, HBcrAg) (see lines 109-121, 132).
  1. Add a space between myr-preS1 and 2-48 in myr-preS12-48 (line 194)
  • We agree with the editor and corrected the text accordingly (see line 206).
  1. Check the spelling of N-myristoylation (line 222).
  • We agree with the editor and corrected the text accordingly (see line 181 and 234)
  1. Double rolling circle replication converting genome to antigenome to genome was not clearly shown in Figure 2
  • According to the editor´s suggestion with modified figure 2. We visualized the rolling circle mechanism in detail. In particular, the conversion from gRNA to agRNA back to gRNA. Moreover, we modified the overall replication cycle for better understanding. The figure legend was also rewritten to match the figure (see figure 2).
  1. PMT, post-modification target or PTM (lines 437, 459, 468, 513 etc.)
  • We corrected all false abbreviations from PMT to PTM and explained the abbreviation once (see line 437, 450, 472, 481, 523).
  1. Correct the typo of Lonafarnib (line 627).
  • We concur with the editor and corrected the text accordingly (see line 634).
  1. BCP instead of BPC (line 689)
  • We agree with the editor and corrected the text accordingly (see line 699).
  1. Lines 912-913, it was Suárez-Amarán et al who developed AAV-HDV1.2 transduced mice referring to [293]. Usai and HER colleagues followed the study [294]
  • We agree with the editor and rephrased the sentence accordingly (see line 922 -924).
